# Amorphous Solid Dispersion Tablets Overcome Acalabrutinib pH Effect in Dogs

**DOI:** 10.3390/pharmaceutics13040557

**Published:** 2021-04-15

**Authors:** Deanna M. Mudie, Aaron M. Stewart, Jesus A. Rosales, Nishant Biswas, Molly S. Adam, Adam Smith, Christopher D. Craig, Michael M. Morgen, David T. Vodak

**Affiliations:** 1Global Research & Development, Lonza, Bend, OR 97703, USA; aaron.stewart@lonza.com (A.M.S.); rosaleja@uw.edu (J.A.R.); nishantbb@gmail.com (N.B.); molly.adam@lonza.com (M.S.A.); adam.smith@lonza.com (A.S.); chris.craig@lonza.com (C.D.C.); michael.morgen@lonza.com (M.M.M.); david.vodak@lonza.com (D.T.V.); 2Pharmaceutics Department, University of Washington, Seattle, WA 98195, USA

**Keywords:** acalabrutinib, amorphous solid dispersion, acid-reducing agent, bioavailability enhancement, kinase inhibitor, pH effect, proton pump inhibitor, spray drying

## Abstract

Calquence^®^ (crystalline acalabrutinib), a commercially marketed tyrosine kinase inhibitor (TKI), exhibits significantly reduced oral exposure when taken with acid-reducing agents (ARAs) due to the low solubility of the weakly basic drug at elevated gastric pH. These drug–drug interactions (DDIs) negatively impact patient treatment and quality of life due to the strict dosing regimens required. In this study, reduced plasma drug exposure at high gastric pH was overcome using a spray-dried amorphous solid dispersion (ASD) comprising 50% acalabrutinib and 50% hydroxypropyl methylcellulose acetate succinate (HPMCAS, H grade) formulated as an immediate-release (IR) tablet. ASD tablets achieved similar area under the plasma drug concentration–time curve (AUC) at low and high gastric pH and outperformed Calquence capsules 2.4-fold at high gastric pH in beagle dogs. In vitro multicompartment dissolution testing conducted a priori to the in vivo study successfully predicted the improved formulation performance. In addition, ASD tablets were 60% smaller than Calquence capsules and demonstrated good laboratory-scale manufacturability, physical stability, and chemical stability. ASD dosage forms are attractive for improving patient compliance and the efficacy of acalabrutinib and other weakly basic drugs that have pH-dependent absorption.

## 1. Introduction

Calquence^®^ is a commercially marketed kinase inhibitor used as second-line therapy for adult patients seeking treatment for mantle cell lymphoma, chronic lymphocytic leukemia, or small lymphocytic lymphoma [1]. The active pharmaceutical ingredient (API) in Calquence is the crystalline Form I of acalabrutinib, which is a diprotic weak base with pK_a_ values of 3.5 and 5.8 [2]. It is a Biopharmaceutics Classification System (BCS) Class 2 drug with low intrinsic solubility and a moderate log P [2,3]. The chemical structure and compound properties of acalabrutinib are shown in Figure 1 and Table 1, respectively.

Calquence has clinically meaningful drug–drug interactions (DDIs) with acid-reducing agents (ARAs) such as proton pump inhibitors (PPIs), where concomitant use results in reduced area under the plasma drug concentration–time curve (AUC) values [4]. For example, coadministration of Calquence with 40 mg of the PPI omeprazole for 5 days decreased AUC by 43% in healthy subjects [1]. According to the Food and Drug Administration (FDA) label for Calquence, patients must avoid coadministration with PPIs and stagger dosing with histamine H_2_ receptor antagonists (H_2_RAs) and antacids [1]. Because ARAs are commonly prescribed to cancer patients, these DDIs can undermine efficacy and patient compliance due to the complex dosing schedules required [5]. 

The mechanism of reduced performance of acalabrutinib crystalline Form I when taken with ARAs is decreased solubility at the elevated gastric pH levels that occur when ARAs are administered, referred to here as the “ARA effect” [4,6]. The reduced solubility of the weakly basic drug decreases dissolution, which decreases absorption across the intestinal membrane and increases the extent of dose metabolized in humans [2,7]. This gastric pH-dependent mechanism that occurs with weakly basic drugs is the most common DDI for ARAs. However, natural variations in gastric pH also cause undesirable variability in Calquence performance, irrespective of the use of ARAs [4,8,9,10].

To overcome the ARA effect, a spray-dried amorphous solid dispersion (ASD) was developed and formulated as an immediate-release (IR) tablet. ASD technologies are used to enhance solubilization and increase the bioavailability of poorly soluble drugs [12]. The amorphous drug form has higher solubility than the crystalline drug form due to its higher activity [13]. Since the amorphous form is thermodynamically unstable, it is typically formulated as an ASD with a polymer to stabilize the high free energy state [14].

Since the ARA effect is caused by the decreased solubility of acalabrutinib crystalline Form I at elevated gastric pH, it was hypothesized that an ASD of acalabrutinib could increase API solubility enough to provide rapid and high extent of gastrointestinal (GI) dissolution of the API at high and low gastric pH values (i.e., in the presence and absence of ARAs). Furthermore, achieving complete absorption in the presence of ARAs would provide limited potential for increased absorption at low, fasted-state gastric pH where the API is more soluble. Achieving similar AUC values under both conditions could thereby mitigate the ARA effect and potentially negate the DDI seen with crystalline Form I acalabrutinib.

The goal of this study was to demonstrate mitigation of the ARA effect in a beagle dog model using acalabrutinib ASD IR tablets at a 100 mg dose. A 50/50 (% *w*/*w*) ASD containing acalabrutinib and hydroxypropyl methylcellulose acetate succinate (HPMCAS, HF grade) was prepared using a laboratory (kilogram)-scale spray dryer and incorporated into ASD IR tablets. In vivo tests were conducted in fasted beagle dogs (1) pretreated with pentagastrin to increase gastric acid secretion and lower stomach pH or (2) pretreated with famotidine (an ARA) to decrease gastric acid secretion and elevate stomach pH. ASD IR tablets and commercially available Calquence capsules were administered to the dogs and the results were compared.

In addition to the in vivo study, ASD tablets and Calquence capsules were tested and compared in an in vitro multicompartment dissolution apparatus designed to reflect average gastrointestinal physiology in beagle dogs pretreated with pentagastrin or famotidine. Further, physical and chemical stability of the ASD, and chemical stability of the ASD tablet were assessed to demonstrate the integrity of amorphous acalabrutinib in the ASD tablet.

## 2. Materials and Methods

### 2.1. Material Sourcing

Acalabrutinib (CAS 1420477-60-6, >98% purity) was purchased from LC Laboratories (Woburn, MA, USA). HPMCAS-HF (Aqoat, HF grade) was purchased from Shin-Etsu Chemical Co., Ltd. (Tokyo, Japan). Hydrochloric acid (HCl), sodium acetate, sodium phosphate, potassium phosphate, and sodium chloride (NaCl) were purchased from Sigma Aldrich Chemical Company (St. Louis, MO, USA). Ammonium acetate was purchased from Thermo Fisher Scientific (Waltham, MA, USA). Fasted-state simulated intestinal fluid (FaSSIF) powder was purchased from Biorelevant.com Ltd. (London, UK). Methanol (HPLC grade) was purchased from Honeywell (Morris Plains, NJ, USA). Calquence capsules were purchased from Drug World (Cold Spring, NY, USA). Avicel PH-101 (microcrystalline cellulose) was purchased from FMC Corporation (Philadelphia, PA, USA). Pearlitol 25 (mannitol) was purchased from Roquette America (Geneva, IL, USA). Ac-Di-Sol (croscarmellose sodium) was purchased from Dupont (Wilmington, DE, USA). Cab-O-Sil M5P (fumed silica) was purchased from Cabot Corporation (Alpharetta, GA, USA). Magnesium stearate was purchased from Macron Fine Chemicals/Avantor (Radnor, PA, USA). Covance Laboratories (Madison, WI, USA) purchased pentagastrin from Sigma Aldrich Chemical Company and famotidine from Kirkland Signature (Kirkland, WA, USA) for use in in vivo testing.

### 2.2. ASD Manufacturing and Characterization

A 32 g batch of 50/50 (% *w*/*w*) acalabrutinib/HPMCAS-HF grade (Aqoat^®^, Shin-Etsu Chemical Co., Ltd., Tokyo, Japan) ASD was manufactured using a customized laboratory-scale spray dryer (0.3 m chamber diameter) capable of drying-gas flow rates of up to 35 kg/h. This ASD composition was selected as the lead after screening several different dispersion polymers and one additional drug loading (data to be provided in a future publication). Methanol was used as the spray solvent. The total solids loading in the spray solution was 5.4 wt%. Solutions were sprayed using a Schlick 2.0 pressure-swirl nozzle (Model 121, 200-μm orifice, Schlick Americas, Bluffton, SC, USA) at an outlet temperature of 45 to 50 °C and an inlet temperature of 142 to 150 °C.

After material was collected in a cyclone, residual solvent was removed in a secondary drying step using a vacuum dryer (Model TVO-2, Cascade TEK, Cornelius, OR, USA) for 20 h at 40 °C with a nitrogen sweep gas (−60 cmHg, 3 standard liters per minute). Solvent removal was confirmed to be below International Council for Harmonization (ICH) thresholds for methanol (<3000 ppm) using gas chromatography (GC). The ASD was analyzed using modulated differential scanning calorimetry (mDSC) to ensure a single T_g_ as an indication of drug-polymer homogeneity. Powder X-ray diffraction (PXRD) was used to verify the ASD was amorphous. Scanning electron microscopy (SEM) was used to visually identify potential changes in morphology or the presence of surface crystals. Method details can be found in Section A.1.

### 2.3. ASD Tablet Manufacturing and Characterization

The ASD IR tablets, which had a 100 mg unit dosage strength, a 400 mg total mass and a drug loading of 25 wt%, were manufactured using the 50/50 (% *w*/*w*) acalabrutinib/HPMCAS-H ASD. ASD tablet information is provided in Table 2.

ASD tablets were manufactured using a small-scale, semi-manual dry-granulation process. The ASD and intragranular excipients (see Table 2) were blended in a Turbula blender (Glen Mills Inc., Clifton, NJ, USA). The intragranular blend was compressed into slugs using a Manesty F3 single-station tablet press (Manesty Ltd., Knowsley, UK) with half-inch flat-faced tooling. Slugs were milled using a 1Zpresso Pro coffee grinder (1Zpresso, New Taipei City, Taiwan) at a setting of 4.5. The milled slugs were blended with extragranular excipients (see Table 2) to create a final blend. The final blend was compressed to a target tensile strength of 2 MPa using a Manesty F3 single-station tablet press with 11 mm SRC tooling. Complete manufacturing details can be found in Section A.2.

Tablet disintegration rates were determined in HCl (pH 2) and phosphate-buffered saline (pH 6, 67 mM phosphate) in a U.S. Pharmacopeia (USP) disintegration apparatus (ZT-71 disintegration tester, Erweka, Heusenstamm, Germany) using three replicates.

### 2.4. Stability

The ASD was stored at 40 °C/75% relative humidity (RH) to assess physical and chemical stability under accelerated storage conditions. Initial ASD samples and aged ASDs were analyzed for crystallinity using PXRD, thermal properties using mDSC, changes in morphology or presence of surface crystals using SEM, and chemical degradation by reverse-phase high-performance liquid chromatography (RP-HPLC). The ASD and ASD tablets were stored at ambient and refrigerated conditions to assess the chemical stability at nonaccelerated conditions. Initial and aged ASD samples were analyzed for total related substances using RP-HPLC. See Section A.3 for method details.

### 2.5. In Vitro Dissolution Testing

The in vitro dissolution performance of the ASD tablet and the commercially available Calquence capsule was evaluated using a controlled-transfer dissolution (CTD) apparatus containing stomach, duodenum, and jejunum/waste compartments [15,16]. The in vitro testing parameters were selected based on acalabrutinib physicochemical properties; the dose; and the average physiology of fasted beagle dogs (1) pretreated with pentagastrin to increase gastric acid secretion and lower stomach pH and (2) pretreated with famotidine to decrease gastric acid secretion and elevate stomach pH [17,18,19,20,21,22,23]. Due to its basic pK_a_s (3.5 and 5.8), acalabrutinib is expected to dissolve to a high extent in low-pH gastric environments and supersaturate upon emptying into the upper intestine [2]. A CTD test was used to evaluate the dissolution behavior of acalabrutinib in various gastric environments and maintain supersaturation upon emptying into an intestinal compartment. CTD test parameters are summarized in Table 3.

Fiber-optic UV probe detection was used to measure acalabrutinib concentrations in the stomach and duodenum compartments of the CTD apparatus. Before the experiment, unique calibration curves were generated for each UV probe (2 mm path length) by delivering aliquots of a known amount of stock acalabrutinib solution (10 mg/mL acalabrutinib in methanol) to 50 mL of gastric or intestinal medium held at 37 °C. To begin dosing, a single 100 mg ASD tablet or 100 mg Calquence capsule was placed in a hanging basket and submerged in the stomach compartment along with 50 mL water (for a 100 mL total volume in the stomach compartment). This achieves a dose concentration of 1 mg/mL acalabrutinib in the stomach compartment at the onset of the experiment. The ASD tablet was agitated for 1 min and the Calquence capsule was agitated for 10 min before initiating gastric emptying to allow dosage-form disintegration and dispersal. Each compartment was stirred at 150 rpm and held at 37 °C by circulating water through jacketed glass vessels. The dissolution performance in each compartment was monitored using Pion Rainbow™ UV probes (Pion Inc., Billerica, MA, USA) at 360 to 364 nm in the stomach and the duodenum compartments. The apparent concentrations measured consisted of (1) drug dissolved in aqueous medium, (2) drug partitioned into bile-salt micelles (when present) as micelle-bound drug. Drug concentrations were monitored for 90 min after gastric emptying started. All samples were analyzed in duplicate.

### 2.6. Pharmacokinetic (PK) Study

ASD tablets and Calquence capsules were evaluated for in vivo performance in beagle dogs (*n* = 6) at a 100 mg dose strength. The study was conducted by Covance Laboratories Inc. (Madison, WI, USA) in accordance with the protocol, protocol amendment, and Covance standard operating procedures (SOPs). All procedures were in compliance with the Animal Welfare Act Regulations (9 CFR 3).

Calquence capsules were stored at room temperature according to package directions and ASD tablets were stored at 2 to 8 °C prior to dosing. The study design is shown in Table 4. Dogs were pretreated with intramuscular pentagastrin in Phases 1 and 3 to increase gastric acid secretion and lower stomach pH (representing fasted human gastric pH) [17]. Dogs were pretreated with oral famotidine in Phases 2 and 4 to decrease gastric acid creation and raise stomach pH (representing the gastric pH of fasted humans taking ARAs) [17,18].

The purebred, non-naive beagle dogs used in the study weighed 8.9 to 10.7 kg and were 1 to 2 years in age. For each phase, animals were fasted overnight before dose administration. Food was returned to the animals at approximately 4 h after dosing. For each phase, a flush with approximately 10 mL water was administered to facilitate swallowing of the tablet or capsule dosage form.

For each phase, blood (approximately 1 mL) was collected from each animal from a jugular vein into tubes containing K_2_EDTA before dosing at 0.25, 0.5, 1, 2, 4, 8, 12, and 24 h after dosing. Blood was maintained in chilled cryogenic racks before centrifugation to obtain plasma. Centrifugation began within 1 h of collection. Plasma was placed into 96-well tubes with barcode labels. Plasma was maintained on dry ice before storage at approximately −70 °C. 

Acalabrutinib concentrations in dog plasma were measured by liquid chromatography–tandem mass spectrometry (LC–MS/MS) using a Sciex API-5000 system (Framingham, MA, USA) equipped with positive-ionization turbo ion spray. The concentrations of the samples were quantitated using a linear calibration curve that was prepared by diluting a stock solution of acalabrutinib in dimethylformamide at a concentration of 20 µg/mL. The stock solution was diluted to a range of 1–1000 ng/mL in 1:1 acetonitrile:water and spiked with blank protein. An internal standard, consisting of labetalol dissolved at 200 ng/mL in acetonitrile, was used to ensure system suitability. The dog plasma samples were prepared for analysis by precipitating the protein in acetonitrile in a 96-well plate and diluting the supernatant in 1 equivalent of acetonitrile. HPLC analysis was conducted using a 50 mm × 2 mm, 2.5 µm pore size Phenomenex Luna C18(2) high-speed-technology (HST) column (Phenomenex, Torrance, CA, USA) at a flow rate of 0.6 mL/min. The mobile phases consisted of 15 mM ammonium formate in water containing 0.1 % formic acid and 0.1 % formic acid in acetonitrile. During analysis, the column temperature was maintained at 50 °C, whereas the sample compartment was maintained at 5 °C.

Pharmacokinetic parameters including maximum drug plasma concentration (C_max_), time to maximum plasma concentration (T_max_), AUC through 24 h (AUC_0–24_), and the AUC extrapolated to infinity (AUC_0–inf_) were determined for each individual subject using Microsoft Excel (Microsoft Corporation, Seattle, WA, USA). AUC_0-24_ and AUC_0-inf_ were calculated using a linear trapezoidal method. Results of each individual subject were averaged to obtain the reported averages and standard deviations. AUC_0-inf_ ratios (famotidine/pentagastrin) were calculated for ASD tablets and Calquence capsules. In addition, the AUC_0-inf_ values for ASD tablets and Calquence capsules were compared with those of Calquence capsules after pentagastrin pretreatment, as were statistical *p*-values using a one-way analysis of variance (ANOVA) test for AUC relative to Calquence after famotidine pretreatment and Calquence after pentagastrin pretreatment.

## 3. Results

### 3.1. ASD and Tablet Manufacturing and Characterization

A 50/50 (% *w*/*w*) acalabrutinib/HPMCAS-H ASD was spray dried with a high yield (97% after secondary drying). The ASD contained 100 ppm of residual methanol, well below the ICH threshold of 3000 ppm. Furthermore, the ASD was confirmed to be amorphous and homogenous by mDSC as evident by a single T_g_ centered at 115.6 ± 0.3 °C. PXRD showed no evidence of crystallinity, based on the absence of sharp diffraction peaks characteristic of crystalline acalabrutinib. SEM demonstrated that ASD particles were primarily collapsed spheres with no evidence of surface crystals. Complete characterization results can be found in Section A.4.

ASD tablets were successfully manufactured to a target tensile strength of 2 MPa. The compression pressure needed to achieve a 2 MPa tensile strength was 78 MPa, with an average solid fraction of 0.80. On average, the tablets were 4.9 mm thick and had a tablet volume of 370 mm^3^. This tablet volume is 60% smaller than the dosage form size of Calquence capsules, which have a calculated Size 1 capsule volume of ~870 mm^3^.

Tablets disintegrated rapidly in the USP disintegration tester, with a 32 ± 2 s disintegration time (average ± standard deviation) at pH 2 and a 37 ± 4 s disintegration time at pH 6.

### 3.2. Stability

The ASD was stored at 40 °C/75% RH to assess physical and chemical stability when stored under accelerated conditions. The ASD showed good physical stability when stored at 40 °C/75% RH/open for 6 months, as indicated by a lack of sharp diffraction peaks by PXRD, a lack of particle fusion or other particle surface morphology changes by SEM micrographs, and a single T_g_ by mDSC after aging at accelerated conditions. Although the ASD showed good physical stability at 40 °C/75% RH, significant chemical degradation of the ASD was observed after storage for 12 weeks at 40 °C/75% RH, with HPLC results showing a 780% increase in impurities.

The ASD and ASD tablet were stored at ambient and refrigerated conditions to assess chemical stability. Storage at refrigerated conditions with and without desiccant and at 25 °C/60% RH/sealed with desiccant mitigated chemical degradation of the ASD and ASD tablet. For example, refrigerated storage for 12 weeks resulted in no increase in impurities for the ASD or ASD tablet. Storage at 25 °C/60% RH/sealed with desiccant for 12 weeks resulted in no increase in impurities for the ASD and only a 6% increase in impurities (total impurities = 0.69 ± 0.00 area%) for the ASD tablet. Results and discussion can be found in Section A.5.

### 3.3. Tablet In Vitro Dissolution Performance

ASD tablets and Calquence capsules were tested in the CTD apparatus using 0.01 N HCl (pH 2) as the gastric medium. As the dissolution results in Figure 2 show, dissolution of the ASD tablets and Calquence capsules were practically identical under conditions simulating fasted dogs treated with pentagastrin. Acalabrutinib is a weakly basic compound with pK_a_s of 3.5 and 5.8. As a result of ionization, acalabrutinib has a high crystalline solubility at pH 2. This high crystalline drug solubility drives rapid dissolution of both dosage forms in the gastric compartment and, upon transfer to the duodenal compartment, results in supersaturated drug concentrations and identical duodenal AUC values.

The dissolution profiles from the CTD apparatus using 1 × 10^−6^ N HCl (pH 6) as the gastric medium are shown in Figure 3. In contrast to the pH 2 results, the ASD tablet outperformed the Calquence capsule in conditions that simulated fasted dogs treated with ARAs such as famotidine. The crystalline solubility of acalabrutinib is orders of magnitude lower at pH 6 than at pH 2. As such, the crystalline drug in the Calquence capsules is solubility-limited in the gastric compartment and does not supersaturate to the same extent when transferred to the duodenal compartment. In contrast, the ASD supersaturates in both the gastric and duodenal compartments, resulting in a 3.4-fold calculated enhancement in duodenal AUC for the ASD tablet relative to the Calquence capsules, as shown in Table 5.

### 3.4. PK Study

ASD tablets and Calquence capsules were dosed to dogs pretreated with pentagastrin or famotidine. Profiles for blood plasma concentration versus time for each formulation treatment from 0 to 12 h are plotted in Figure 4, with tabulated noncompartmental PK results in Table 6. Profiles of blood plasma drug concentration versus time from 0 to 24 h and additional noncompartmental PK results can be found in Section A.6. The performance of the Calquence capsules and ASD tablets was similar after pentagastrin pretreatment. This result is in line with expectations prior to study initiation and supports historical performance data in dogs and humans, where acalabrutinib is well absorbed when administered to subjects with low stomach pH [2,6].

For the famotidine pretreatment condition, the ASD tablet achieved a 2.4-fold higher AUC than the Calquence capsules, as well as approximately 93% of the AUC observed for the pentagastrin-treated dogs, overcoming the impact of stomach pH. As expected, the performance of the Calquence capsules suffered at high stomach pH, resulting in roughly a 3-fold decrease in AUC_0-inf_ compared to the pentagastrin-treated dogs. Again, this is in line with expectations based on historical performance data for Calquence dosed with ARAs. The AUC_0-inf_ values for the ASD tablets in pentagastrin- and famotidine-treated dogs were statistically equivalent to each other, and to that of the Calquence capsules for pentagastrin-treated dogs. However, the AUC_0-inf_ values for the ASD tablet in pentagastrin- and famotidine-treated dogs were statistically higher than the AUC_0-inf_ values for Calquence capsules in famotidine-treated dogs.

## 4. Discussion

### 4.1. ASDs for Improving Low-Solubility, Weakly Basic Drugs

ASD dosage forms are attractive for enhancing the oral exposure of poorly soluble (e.g., BCS Class 2 and 4) drugs. The ASD tablets in this study boost AUC values 2.4-fold compared to that of Calquence capsules in beagle dogs at high gastric pH levels. Based on this result, ASD tablets are a promising alternative to Calquence capsules for improving patient compliance and efficacy [5]. Currently, patients are instructed to avoid co-administration with PPIs and to stagger dosing with H_2_RAs and antacids, so these ASD tablets present the potential for co-administration with all types of ARAs [1]. Furthermore, these ASD tablets are 60% smaller than Calquence capsules at the same unit dosage strength, which should make them easier for patients to swallow [24,25]. While the ASD tablets were manufactured with a round, convex shape for this dog study, the size reduction is expected to translate to other tablet shapes (e.g., oblong) that may further facilitate swallowability in humans [24].

The results of this study are generally applicable to other small-molecule protein kinase inhibitors besides acalabrutinib, as well as weakly basic drugs in general. Several weakly basic, oral oncologic drugs on the market have shown evidence of decreased absorption as a result of high gastric pH when taken with ARAs [26,27]. Weakly basic drugs made up 78% of new molecular entities approved between 2003 and 2013 that showed a clinical DDI with ARAs [4]. Since small-molecule protein kinase inhibitors are prevalent in pharmaceutical pipelines and many are poorly soluble, ASD dosage forms have the potential to enable effective delivery and improve the experience of many cancer patients [28,29,30,31]. In addition to removing the ARA effect as highlighted in this study, ASD dosage forms can also mitigate food effects, decrease plasma variability and reduce dose [30,32].

### 4.2. In Vitro Versus In Vivo Performance

The in vitro CTD methodology forecast relative in vivo performance between the ASD tablet and Calquence capsules a priori to the dog study. As shown in Figure 5, the in vivo AUC values of ASD tablets relative to that of Calquence capsules in the pentagastrin pretreatment phase showed the same rank-ordering as in vitro duodenal AUC values relative to those of Calquence capsules at pH 2 (the initial gastric pH condition).

On average, the ASD tablets performed somewhat better in vivo than they did in vitro relative to the Calquence capsules at low initial gastric pH. This result is not surprising, because high-permeability drugs such as acalabrutinib often perform better in vivo than in vitro using dissolution test apparatuses that lack an absorption compartment [33,34]. Coupling in vitro testing with in silico modeling is a useful strategy for capturing the sensitivity of formulations to the multitude of important physiological variables [35,36]. Nonetheless, in this study, the CTD methodology proved useful in forecasting the in vivo performance of the ASD tablets and Calquence capsules.

### 4.3. Dog Model for Studying ARA Effect

In this study, a dog model was chosen to demonstrate the ability of an ASD dosage form to mitigate the acalabrutinib ARA effect. Beagle dogs are a common preclinical species and surrogate for humans due to their ability to swallow intact dosage forms and the relative ease in maintaining the animals. Although beagle dogs can be useful surrogates for humans, several differences exist between human and beagle dog physiology, including gastric pH [23,37,38].

It is common practice to modulate gastric pH in dogs when evaluating oral drug product performance. Pretreatment with intramuscular pentagastrin or oral famotidine has provided gastric pH values in beagle dogs that were more consistent and stable than those of dogs that had not been pretreated—in the range of pH 1–2 (for pentagastrin) or pH 7 (for famotidine) [17,18]. These findings make intramuscular pentagastrin and oral famotidine pretreatments useful for studying the impact of a wide range in gastric pH on weakly basic drugs, as was done in this study [17,18,39,40,41].

Famotidine represents a worst-case scenario for weakly basic drugs since a pH of 7 falls at the high end of the pH range, where drug solubility is lowest. Measured gastric pH in humans after taking ARAs can range from approximately pH 3 to 7 depending upon type of ARA, dose, duration of treatment, and individual response [42,43,44]. Therefore, the fact that the difference in AUC of the ASD tablet was only 7% over such a wide pH range (i.e., ~1 to 7), suggests low pH sensitivity and high-performance robustness for different ARA treatments. Further, it suggests high robustness of the ASD tablet to the natural pH variations that can be found even in patients who are not taking ARAs [8].

In attempting to use the current dog study results to forecast the utility of ASDs for overcoming pH effects in humans, it should be noted that acalabrutinib metabolism differs substantially between humans and dogs. In humans, approximately 50% of acalabrutinib is metabolized by CYP3A4 in the gut and 50% is metabolized by first-pass liver extraction [2]. In contrast, in dogs, acalabrutinib undergoes less-extensive metabolism upon first pass through the gut and liver, with reports of bioavailability in the 70% to 80% range at the human prescribed dose (100 mg) [2,45]. In humans, lower solubility at high gastric pH purportedly reduces absorption across the GI membrane and increases the extent of metabolized drug, whereas in dogs, the lower solubility at higher pH is expected to primarily reduce absorption [2]. Despite these metabolism differences, in both dogs and humans, the ASD tablet increases dissolved drug concentrations at elevated gastric pH levels. Therefore, success in dogs suggests that the ASD tablets will also overcome the ARA effect in humans.

## 5. Conclusions

This study demonstrates that ASD tablets are an effective enabling technology for overcoming reduction in AUC of the weakly basic drug acalabrutinib when co-administered with a gastric ARA. In beagle dogs, ASD tablets achieved similar AUC values at low and high gastric pH conditions and outperformed Calquence capsules 2.4-fold at high gastric pH. Relative formulation performance was successfully forecast using a multicompartment in vitro CTD apparatus. 

The ASD was easy to manufacture at high yield on laboratory-scale spray-drying equipment and the resulting ASD tablets were 60% smaller than Calquence capsules. The ASD had good physical stability and the ASD and ASD tablet showed good chemical stability when stored refrigerated or at room temperature with a desiccant. An ASD dosage form represents a useful strategy for improving patient compliance and efficacy of acalabrutinib. This strategy could be extended to other small-molecule drug products and problem statements requiring bioavailability enhancement to drive improved in vivo performance. 

## Figures and Tables

**Figure 1 pharmaceutics-13-00557-f001:**
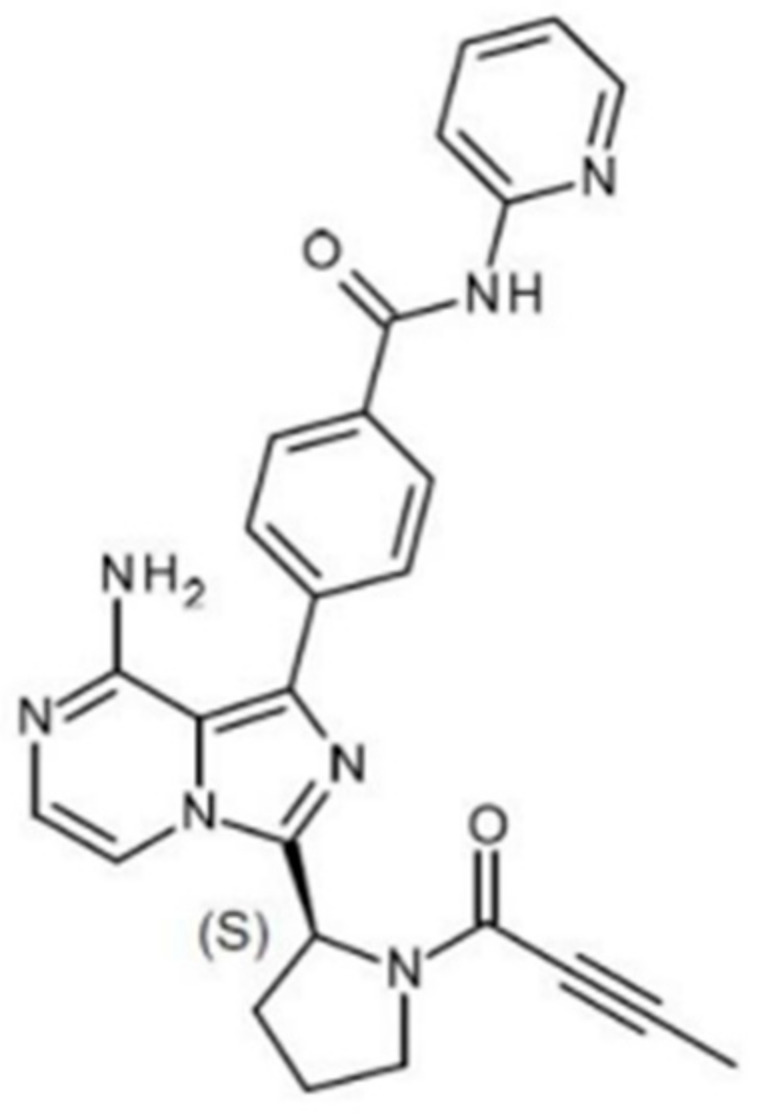
Acalabrutinib chemical structure.

**Figure 2 pharmaceutics-13-00557-f002:**
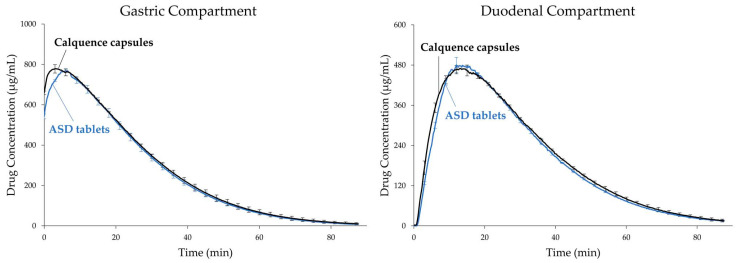
Dissolution profiles of acalabrutinib ASD tablets and Calquence capsules in CTD apparatus, using pH 2 gastric conditions to simulate fasted dogs treated with pentagastrin (*n* = 2).

**Figure 3 pharmaceutics-13-00557-f003:**
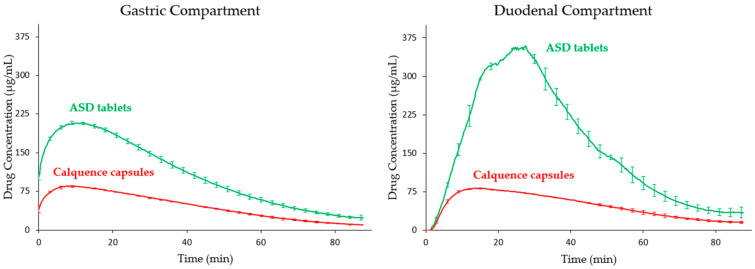
Dissolution profiles of acalabrutinib ASD tablets and Calquence capsules in the CTD apparatus under pH 6 gastric conditions to simulate fasted dogs treated with famotidine (ARAs) (*n* = 2).

**Figure 4 pharmaceutics-13-00557-f004:**
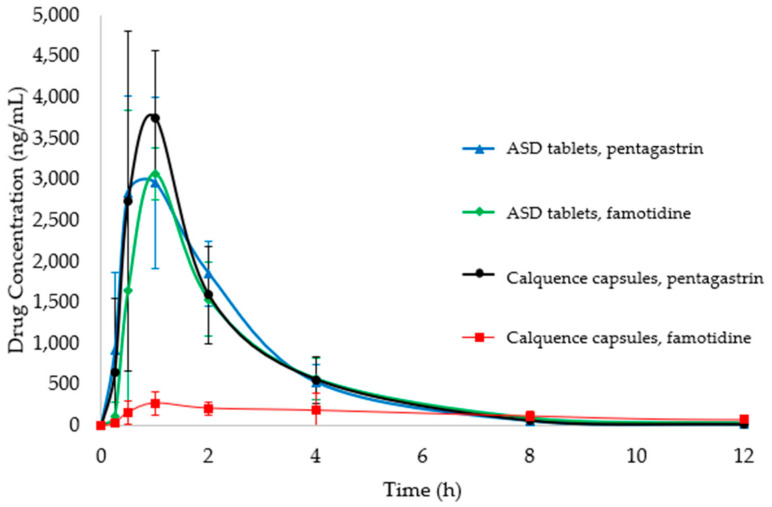
Profiles for blood plasma concentration versus time from 0 to 12 h measured in beagle dogs for ASD tablets and Calquence capsules (*n* = 6).

**Figure 5 pharmaceutics-13-00557-f005:**
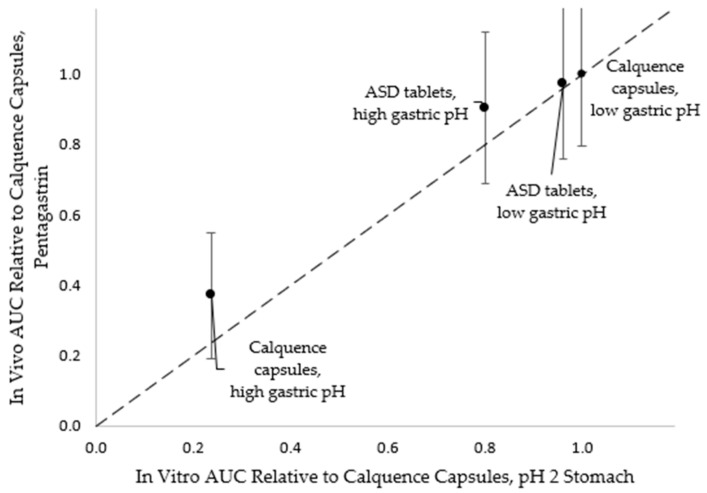
AUC values relative to those of Calquence capsules (pentagastrin pretreatment) in vivo versus duodenal AUC values relative to those of Calquence capsules (pH 2 initial gastric pH) in vitro. Relative AUC is calculated using average AUC. Error bars represent fractional uncertainties, where uncertainty in AUC is the standard deviation. Dashed line = 1:1 correlation line.

**Table 1 pharmaceutics-13-00557-t001:** Acalabrutinib Form I physicochemical properties.

Compound Property	Value
Molecular weight (g/mol)	465.5
pK_a_s in physiological range	3.5 (basic) and 5.8 (basic) ^a^
log P	2.0 ^a^
Melting temperature (T_m_) (°C)	214 ^b^
Glass-transition temperature (T_g_) (°C)	133 ^c^
Crystalline intrinsic solubility (µg/mL)	48 ^a^

^a^ Reference [2]. ^b^ Reference [11]. ^c^ Measured in house using modulated differential scanning calorimetry (mDSC). Refer to Section A.1 for method information.

**Table 2 pharmaceutics-13-00557-t002:** Detailed ASD tablet formulation and composition.

Function	Ingredient	Tablet Fraction (wt%) ^a^
**Intragranular**		
ASD	50/50 acalabrutinib/Aqoat (HPMCAS-HF)	50.0
Ductile filler	Avicel^®^ PH-101	14.3
Brittle filler	Pearlitol^®^ 25	28.7
Disintegrant	Ac-Di-Sol^®^	6.0
Glidant	Cab-O-Sil^®^ (M5P)	0.25
Lubricant	Magnesium stearate	0.25
**Extragranular**		
Glidant	Cab-O-Sil	0.25
Lubricant	Magnesium stearate	0.25
	Total tablet mass:	400 mg

^a^ Tablets were made using 11 mm standard round concave (SRC) tooling.

**Table 3 pharmaceutics-13-00557-t003:** Summary of CTD in vitro testing parameters.

Parameter	Value
Dose (mg)	100
Dosing volume (mL)	50
Dosing medium	Milli-Q water
Gastric resting mediumGastric secretion medium	HCl (pH 2) and 34 mM NaClHCl (pH 6) and 34 mM NaCl
Gastric resting volume (mL)	50
Gastric secretion rate (mL/min)	2.4
Gastric emptying half-life (monoexponential) (min)	15
Duodenal resting and secretion medium	Phosphate (pH 6.5) and FaSSIF powder ^a^
Duodenal fluid volume (mL)	50
Duodenal secretion rate (mL/min)	2.4
Duodenal emptying rate	Gastric emptying and duodenal secretion
Jejunal medium	Gastric and duodenal composition
Jejunal volume	Starts at 0 and increases to 419 mL at 90 min

^a^ Consisted of 67 mM phosphate-buffered saline containing 1% (*w*/*w*) (13.4 mM) FaSSIF powder and 82 mM NaCl.

**Table 4 pharmaceutics-13-00557-t004:** In vivo study design.

Phase ^a^	Test Article	Pretreatment	Dosage Form per Animal
1	Acalabrutinib ASD	Pentagastrin ^b^	1 tablet
2	Acalabrutinib ASD	Famotidine ^c^	1 tablet
3	Calquence	Pentagastrin ^b^	1 capsule
4	Calquence	Famotidine ^c^	1 capsule

^a^ Each phase consisted of 6 dogs at a 100 mg target acalabrutinib dose level with a 7 day washout between phases. ^b^ For Phases 1 and 3, animals received a subcutaneous injection of pentagastrin (6 mg/kg/60 mg/mL/0.1 mL/kg) approximately 30 min before test article administration. ^c^ For Phases 2 and 4, animals received a 40 mg oral dose of famotidine (two 20 mg tablets) approximately 60 min before test article administration.

**Table 5 pharmaceutics-13-00557-t005:** Calculated in vitro AUC in the duodenal compartment during CTD testing of ASD tablets and Calquence capsules.

Test Article with Simulated Pretreatment	Average In Vitro Duodenal AUC (µg-min/mL)	In Vitro AUC Ratio (Famotidine/Pentagastrin)
ASD tablet, pentagastrin (pH 2 stomach)	16,946	0.83
ASD tablet, famotidine (pH 6 stomach)	14,102	-
Calquence, pentagastrin (pH 2 stomach)	17,617	0.24
Calquence, famotidine (pH 6 stomach)	4173	-

**Table 6 pharmaceutics-13-00557-t006:** Noncompartmental PK results from the acalabrutinib beagle dog study. Data are reported as the mean with the standard deviation in parentheses.

Test Article, Pretreatment	ASD Tablet, Pentagastrin	ASD Tablet, Famotidine	Calquence Capsule, Pentagastrin	Calquence Capsule, Famotidine
AUC_0-inf_ (ng-h/mL)	8161 (1364) ^a^	7579 (1423) ^a^	8365 (1201)	3112 (1415) ^b^
C_max_ (ng/mL)	3332 (769)	3443 (996)	4480 (516)	355 (121)
T_max_ (h)	0.9 (0.5)	0.9 (0.2)	0.8 (0.2)	1.6 (1.2)
AUC ratio compared to Calquence capsule, pentagastrin	0.98	0.91	1.00	0.37

^a^ Statistically equivalent to Calquence capsule, pentagastrin (*p* > 0.05). ^b^ Statistically different from Calquence capsule, pentagastrin (*p* < 0.05).

## Data Availability

The data in this study in the form of Microsoft Excel worksheets are available from the corresponding author upon request.

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
