# Peer review of "Amorphous Solid Dispersion Tablets Overcome Acalabrutinib pH Effect in Dogs"

_pharmaceutics, 2021, doi:10.3390/pharmaceutics13040557_

Round 1

Reviewer 1 Report

Hello Authors,

The manuscript studies about the pH dependent drug release of acalabrutinib and how the ASD technology was employed to address this issue. Both in-vitro and in-vivo studies were done along with physicochemical characteristics.

Please see my comments,

1) Shouldn't this be pH 1.2? Tablet disintegration rates were determined in HCl (pH 2), Gastric resting medium HCl (pH 2).

2) Figure A3 should have the pure drug, HPMCAS as reference in the thermograms.

3) What impurities were tested during stability?

4) Was the content uniformity (for API) in the tablets measured? Was it also studied during stability? How are the authors confident that each tablet had the right dosage of acalabrutinib?

Author Response

Reviewer #1:

Hello Authors,

The manuscript studies about the pH dependent drug release of acalabrutinib and how the ASD technology was employed to address this issue. Both in-vitro and in-vivo studies were done along with physicochemical characteristics.

Thank-you for your review. Note, references to line numbers reflect the line number when document is viewed in “simple markup” or “no markup” mode. Line numbers change when document is viewed in “all markup” mode.

Please see my comments,

1) Shouldn't this be pH 1.2? Tablet disintegration rates were determined in HCl (pH 2), Gastric resting medium HCl (pH 2).

Thanks for your review. We indeed determined disintegration rates at pH 2 (not at pH 1.2). This pH value matched the pH value we used in the gastric compartment in the ‘low pH’ dissolution testing, and is line with the reported pH range of 1-2 in beagle dogs treated with pentagastrin, according to references 17 and 18 in the manuscript.

2) Figure A3 should have the pure drug, HPMCAS as reference in the thermograms.

We have added a figure (panel (b) of Figure A3) that shows API, HPMCAS-H and ASD thermograms on the same plot.

3) What impurities were tested during stability?

Studying the amount of total impurities rather than identifying each impurity was the main focus of our chemical stability analysis. However, we did take the first step towards identifying impurities using reverse phase high performance liquid chromatography combined with mass spectrometry (RP-HPLC-MS). The table below summarizes proposed degradants based upon this method. Confidently identifying each degradant would require additional experimentation, which was outside the scope of this study. The degradant that increased the most upon storage was Ad4, with the proposed degradation pathway being alkyne hydration.

We have added the RP-HPLC-MS methodology to section 1 of Appendix A. We have also added the table below and the following text to section 5 of Appendix A, starting on line 169.

“Quantifying total impurities rather than identifying each impurity was the main focus of the chemical stability analysis. However, RP-HPLC-MS analysis provided a first step toward identifying impurities, as shown in Table A4. The degradant that increased the most upon storage was Ad4, with the proposed degradation pathway being alkyne hydration. Confidently identifying each degradant would require additional experimentation, which was outside the scope of this study.”

Peak Retention Time (min)​

Relative Retention Time (min)

Degradant

Characteristic Ion mass-to-charge ratio (m/z)​

Proposed Degradation​ Pathway

Formula Weight of Proposed Product (Da)​a

11.05 â€‹

0.65

Ad1

442.6​

Transamidation with acetate at pyrrolidine ring

​

441.4​

14.79​

0.87

Ad2

400.6​

Amide cleavage at  pyrrolidine ring ​

399.4​

15.09​

0.89

Ad3

550.8​

Putative addition product​

​

16.87​

0.99

Ad4

484.6​

Alkyne hydration

483.5​

17.45​

1.03

Ad5

532.7​

Amidation of free amine with 2-butyne-4-one fragment

532​

20.23​

1.19

Ad6

546 / 548​

Succinamide formation at free amine

548​

23.07​

1.36

Ad7

457.7​

​Unknown

​

25.8​

1.52

Ad8

932.2 / 466.7​

Addition and  dimerization

​

27.9​

1.64

Ad9

888/444​

​Dimerization

​

a Formula weight is typically ~ 1 unit lower than the characteristic ion mass-to-charge ratio (m/z) because each ion picks up an extra H+ during ionization.

4) Was the content uniformity (for API) in the tablets measured? Was it also studied during stability? How are the authors confident that each tablet had the right dosage of acalabrutinib?

We did not measure API content uniformity on the tablets. However, we did assay the tablets prior to and immediately after the in vivo study. Testing results upon tablet manufacturing and shortly after the in vivo study showed 100 – 101 %label claim (n=2) and 102 – 103 % label claim (n=2), respectively. We have added this information to section 5, lines 197-199 of Appendix A.

Reviewer 2 Report

I have reviewed through the whole manuscript critically and I found that authors have attempted a very good work on “Amorphous solid dispersion overcome DDI of acalabrutinib with ARAs”. Below please find my comments:

Line 126, typo (mDSC

Line 200, typo naïve

Line 320, for the mark of green plot in Figure 4, it should be ASD tablets, famotidine. Please double check.

For Appendix A, Figure A2,  it’s better to also provide SEM of 50/50 (wt./wt.) physical mixture of acal/HPMCAS or pure API SEM as the control group.

For an additional question or an open discussion, I am curious whether the authors have ever considered any other grade of HPMCAS, like M grade?

Author Response

Reviewer #2:

I have reviewed through the whole manuscript critically and I found that authors have attempted a very good work on “Amorphous solid dispersion overcome DDI of acalabrutinib with ARAs”. Below please find my comments:

Thank-you for your review. Note, references to line numbers reflect the line number when document is viewed in “simple markup” or “no markup” mode. Line numbers change when document is viewed in “all markup” mode.

Line 126, typo (mDSC

Thank-you. We have corrected to read, “(mDSC)”

Line 200, typo naïve

We have modified from “non-naïve” to “non-naive" on line 201.

Line 320, for the mark of green plot in Figure 4, it should be ASD tablets, famotidine. Please double check.

Thank-you. We have corrected Figure 4 such that the green curve is labeled, “ASD tablets, famotidine”.

For Appendix A, Figure A2, it’s better to also provide SEM of 50/50 (wt./wt.) physical mixture of acal/HPMCAS or pure API SEM as the control group.

We have included an SEM of the acalabrutinib purchased from LC Laboratories. Please see Figure A2(b) in Appendix A.

For an additional question or an open discussion, I am curious whether the authors have ever considered any other grade of HPMCAS, like M grade?

Great question. Yes, we did consider other grades of HPMCAS and other polymers. We prepared ASD intermediates using HPMCAS-L, HPMCAS-M and HPMCAS-H as well as some non-enteric polymers (PVP K30, PVP VA64 and HPMC-E3). In vitro dissolution testing using a gastric-to-intestinal transfer test (pH 5 gastric to pH 6.5 intestinal, and pH 6 gastric to pH 6.5 intestinal) showed that ASDs prepared using HPMCAS-H achieved the highest AUC in intestinal medium compared to all other ASDs. We did not include this information in the current manuscript because we plan to write a second manuscript detailing the formulation development process. We decided that including all of the information in one manuscript would make it too long.

We added the following statement on line 109, “This ASD composition was selected as the lead after screening several different dispersion polymers and one additional drug loading (data to be provided in a future publication).”

Reviewer 3 Report

  1. What is the drug content in the prepared formulations.
  2. There is no solid state characterization image performed for pure drug as well as prepared Tablets to compare the change in the crytallinity.
  3. What is the slection criteria to prepare one composition only. Why the authors have not reported the 8-10 different composition and then select the best one to perform animal study.
  4. In animal study four groups taken, femotidine and femotidine with Calquence Capsul. Is it marketed product or in lab prepared sample. There must also the invitro data added in manuscript.

Author Response

Reviewer #3:

Thank-you for your review. Note, references to line numbers reflect the line number when document is viewed in “simple markup” or “no markup” mode. Line numbers change when document is viewed in “all markup” mode.

  1. What is the drug content in the prepared formulations.

Thanks for your comment. We have added reference to the dosage strength in the main text materials & methods section, lines 90-91, stating, “The ASD IR tablets, which had a 100 mg unit dosage strength…). In addition, in Appendix A, section 5, lines 197-199, we have added the following information:

“Tablets were assayed using the RP-HPLC method upon manufacturing and shortly after the in vivo study (upon receipt of retain tablets from the Covance Laboratories study site), achieving 100 – 101 %label claim (n=2) and 102 – 103 % label claim (n=2), respectively.”

  1. There is no solid state characterization image performed for pure drug as well as prepared Tablets to compare the change in the crytallinity.

We have added an SEM image of as-received acalabrutinib API to panel (b) of Figure A2 in Appendix A. We did not create an SEM image of tablets. In our experience, it is difficult to detect crystals visually on tablet surfaces given the crystal like morphology of the additional excipients”.

  1. What is the slection criteria to prepare one composition only. Why the authors have not reported the 8-10 different composition and then select the best one to perform animal study.

Great question. We did consider other polymers (HPMCAS-L, HPMCAS-M, PVP K30, PVP VA64 and HPMC-E3) and also a lower (25%) drug loading, spray-drying 9 different ASD compositions in total. In vitro dissolution testing using a gastric-to-intestinal transfer test (pH 5 gastric to pH 6.5 intestinal, and pH 6 gastric to pH 6.5 intestinal) showed that ASDs prepared using HPMCAS-H achieved the highest AUC in intestinal medium compared to all other ASDs. Also, similar AUCs were achieved using 25% and 50% drug loading HPMCAS-H ASDs, with the 50% drug loading ASD benefiting from a reduction in tablet size. This result prompted us to choose 50/50 acalabrutinib/HPMCAS-H ASD as the lead, with no perceived benefit of taking additional formulations forward into the in vivo study.

We did not include this information in the current manuscript because we plan to write a second manuscript detailing the formulation development process. We decided that including all of the information in one manuscript would make it too long. We added the following statement on line 109, “This ASD composition was selected as the lead after screening several different dispersion polymers and one additional drug loading (data to be provided in a future publication).”

  1. In animal study four groups taken, femotidine and femotidine with Calquence Capsul. Is it marketed product or in lab prepared sample. There must also the invitro data added in manuscript.

Every reference to the Calquence capsule in this manuscript refers to the marketed product. Therefore, phases 3 and 4 detailed in Table 4 in the main text refer to the marketed Calquence capsule. In addition, the in vitro dissolution testing described in the manuscript was performed using the marketed Calquence capsule. Figure 2 shows the Calquence capsule compared to the ASD tablet using an in vitro gastric pH of 2 (to simulate fasted dogs taking pentagastrin) and Figure 3 shows the Calquence capsule compared to the ASD tablet using an in vitro gastric pH of 6 (to simulate fasted dogs taking famotidine/ARAs). In the introduction, line 77, and in the in vitro dissolution testing methods section, line 154, the words “commercially available” were included.

Reviewer 4 Report

The research study by Mudie et al. aimed to overcome the effect of acid-reducing agents on the oral bioavailability of a weekly basic drug i.e. Calquence® using spray-dried amorphous solid dispersion (ASD) approach. The results showed that ASD tablets achieved the same plasma drug concentration both at low and high gastric pH conditions. In general, the study is well designed, written, and easy to follow while the topic covered is interesting to readers. I would strongly recommend accepting this manuscript for publication after a minor revision for the following comments.

Lines 70-86: This part should be deleted as it is not appropriate for the introduction. Delete and update the introduction part with clear objectives of the study.

Line 90:  Delete hyphen between the number and unit i.e., 400-mg. Correct it throughout the manuscript.

Section 2.1. Move and merge this section in 2.4.

Line 128 and 147: Scanning electron microscopy only gives the surface morphology information. Correct it.

Section 3.1. Discuss the surface morphology of the spray-dried powders characterized using SEM.

Section 3.1: Add standard deviation values for the disintegration times.

Line 262: Mention the names of possible impurities.

Author Response

Reviewer #4:

The research study by Mudie et al. aimed to overcome the effect of acid-reducing agents on the oral bioavailability of a weekly basic drug i.e. Calquence® using spray-dried amorphous solid dispersion (ASD) approach. The results showed that ASD tablets achieved the same plasma drug concentration both at low and high gastric pH conditions. In general, the study is well designed, written, and easy to follow while the topic covered is interesting to readers. I would strongly recommend accepting this manuscript for publication after a minor revision for the following comments.

Thank-you for your review. Note, references to line numbers reflect the line number when document is viewed in “simple markup” or “no markup” mode. Line numbers change when document is viewed in “all markup” mode.

Lines 70-86: This part should be deleted as it is not appropriate for the introduction. Delete and update the introduction part with clear objectives of the study.

Thanks for your comment. The section of the introduction that you referenced has been revised as follows:

“The goal of this study was to demonstrate mitigation of the ARA effect in a beagle dog model using acalabrutinib ASD IR tablets at a 100-mg dose. A 50/50 (% w/w) ASD containing acalabrutinib and hydroxypropyl methylcellulose acetate succinate (HPMCAS, HF grade) was prepared using a laboratory (kilogram)-scale spray dryer and incorporated into ASD IR tablets. In vivo tests were conducted in fasted beagle dogs (1) pretreated with pentagastrin to increase gastric acid secretion and lower stomach pH or (2) pretreated with famotidine (an ARA) to decrease gastric acid secretion and elevate stomach pH. ASD IR tablets and commercially available Calquence capsules were administered to the dogs and the results were compared.

In addition to the in vivo study, ASD tablets and Calquence capsules were tested and compared in an in vitro multicompartment dissolution apparatus designed to reflect average gastrointestinal physiology in beagle dogs pretreated with pentagastrin or famotidine. Further, physical and chemical stability of the ASD, and chemical stability of the ASD tablet were assessed to demonstrate the integrity of amorphous acalabrutinib in the ASD tablet.”

Line 90: Delete hyphen between the number and unit i.e., 400-mg. Correct it throughout the manuscript.

We have deleted the hyphen between the number and the unit in all cases in the manuscript.

Section 2.1. Move and merge this section in 2.4.

We have moved the original section 2.1 into section 2.4 (which has now become section 2.3). We have also made the minor edits required to ensure clarity of each section given the change.

Line 128 and 147: Scanning electron microscopy only gives the surface morphology information. Correct it.

The previous line 128 (now line 121) has been changed to state, “Powder X-ray diffraction (PXRD) was used to verify the ASD was amorphous. Scanning electron microscopy (SEM) was used to visually identify potential changes in morphology or the presence of surface crystals.” The previous line 147 (now line 147) has been changed to state, “…were analyzed for crystallinity using PXRD, thermal properties using mDSC, changes in morphology or presence of surface crystals using SEM,…”.

Section 3.1. Discuss the surface morphology of the spray-dried powders characterized using SEM.

Section 3.1, line 244 now states, “SEM demonstrated that ASD particles were primarily collapsed spheres with no evidence of surface crystals.”.

Section 3.1: Add standard deviation values for the disintegration times.

We have added the standard deviations for the disintegration times as follows on line 253, “Tablets disintegrated rapidly in the USP disintegration tester, with a 32 ± 2-second  disintegration time (average ± standard deviation) at pH 2 and a 37± 4-s disintegration time at pH 6.” In doing so we removed the range in disintegration times of the three replicates, as we thought it redundant information.

Line 262: Mention the names of possible impurities.

Studying the amount of total impurities rather than identifying each impurity was the main focus of our chemical stability analysis. However, we did take the first step towards identifying impurities using reverse phase high performance liquid chromatography combined with mass spectrometry (RP-HPLC-MS). The table below summarizes proposed degradants based upon this method. Confidently identifying each degradant would require additional experimentation, which was outside the scope of this study. The degradant that increased the most upon storage was Ad4, with the proposed degradation pathway being alkyne hydration.

We have added the RP-HPLC-MS methodology to section 1 of Appendix A. We have also added the table below and the following text to section 5 of Appendix A, starting on line 169.

“Quantifying total impurities rather than identifying each impurity was the main focus of the chemical stability analysis. However, RP-HPLC-MS analysis provided a first step toward identifying impurities, as shown in Table A4. The degradant that increased the most upon storage was Ad4, with the proposed degradation pathway being alkyne hydration. Confidently identifying each degradant would require additional experimentation, which was outside the scope of this study.”

Peak Retention Time (min)​

Relative Retention Time (min)

Degradant

Characteristic Ion mass-to-charge ratio (m/z)​

Proposed Degradation​ Pathway

Formula Weight of Proposed Product (Da)​a

11.05 â€‹

0.65

Ad1

442.6​

Transamidation with acetate at pyrrolidine ring

​

441.4​

14.79​

0.87

Ad2

400.6​

Amide cleavage at  pyrrolidine ring ​

399.4​

15.09​

0.89

Ad3

550.8​

Putative addition product​

​

16.87​

0.99

Ad4

484.6​

Alkyne hydration

483.5​

17.45​

1.03

Ad5

532.7​

Amidation of free amine with 2-butyne-4-one fragment

532​

20.23​

1.19

Ad6

546 / 548​

Succinamide formation at free amine

548​

23.07​

1.36

Ad7

457.7​

​Unknown

​

25.8​

1.52

Ad8

932.2 / 466.7​

Addition and  dimerization

​

27.9​

1.64

Ad9

888/444​

​Dimerization

​

a Formula weight is typically ~ 1 unit lower than the characteristic ion mass-to-charge ratio (m/z) because each ion picks up an extra H+ during ionization.